# Building an understanding of Ethnic minority people's Service Use Relating to Emergency care for injuries: the BE SURE study protocol

Fadi Baghdadi ,[1] Bridie Angela Evans ,[1] Steve Goodacre ,[2] Paul Anthony John,[3] Thanuja Hettiarachchi,[4] Ann John ,[1] Ronan A Lyons ,[1] Alison Porter ,[1] Solmaz Safari,[4] Aloysius Niroshan Siriwardena ,[5] Helen Snooks,[1] Alan Watkins ,[1] Julia Williams ,[6] Ashrafunnesa Khanom [1]

[1]Medical School, Swansea University, Swansea, UK
[2]School of Health and Health Related Research, University of Sheffield, Sheffield, UK
[3]Research and Innovation Hub, Scottish Ambulance Service, Edinburgh, UK
[4]Public Contributor, c/o Medical School, Swansea University, Swansea, UK
[5]School of Health and Social Care, University of Lincoln, Lincoln, UK
[6]School of Health and Social Work, University of Hertfordshire, Hertfordshire, UK

**Correspondence to**
Dr Fadi Baghdadi;
fadi.baghdadi@swansea.ac.uk

## ABSTRACT

**Introduction** Injuries are a major public health problem which can lead to disability or death. However, little is known about the incidence, presentation, management and outcomes of emergency care for patients with injuries among people from ethnic minorities in the UK. The aim of this study is to investigate what may differ for people from ethnic minorities compared with white British people when presenting with injury to ambulance and Emergency Departments (EDs).

**Methods and analysis** This mixed methods study covers eight services, four ambulance services (three in England and one in Scotland) and four hospital EDs, located within each ambulance service. The study has five Work Packages (WP): (WP1) scoping review comparing mortality by ethnicity of people presenting with injury to emergency services; (WP2) retrospective analysis of linked NHS routine data from patients who present to ambulances or EDs with injury over 5 years (2016–2021); (WP3) postal questionnaire survey of 2000 patients (1000 patients from ethnic minorities and 1000 white British patients) who present with injury to ambulances or EDs including self-reported outcomes (measured by Quality of Care Monitor and Health Related Quality of Life measured by SF-12); (WP4) qualitative interviews with patients from ethnic minorities (n=40) and focus groups—four with asylum seekers and refugees and four with care providers and (WP5) a synthesis of quantitative and qualitative findings.

**Ethics and dissemination** This study received a favourable opinion by the Wales Research Ethics Committee (305391). The Health Research Authority has approved the study and, on advice from the Confidentiality Advisory Group, has supported the use of confidential patient information without consent for anonymised data. Results will be shared with ambulance and ED services, government bodies and third-sector organisations through direct communications summarising scientific conference proceedings and publications.

## STRENGTHS AND LIMITATIONS OF THIS STUDY

⇒ Mixed quantitative and qualitative methods will ensure representativeness and depth in addressing our research questions.
⇒ The study will use peer researchers from ethnic minorities recruited by local third-sector organisations to support people to complete questionnaires to ensure local buy-in and to achieve as high a response rate to the survey as possible.
⇒ Routine linked data allow the inclusion of a large number of patients and attendances over a 5-year period, producing a comprehensible epidemiological picture.
⇒ Coding of ethnicity may be inaccurately recorded or incomplete in routine health records which may mask heterogeneity within each group.
⇒ Response rates to questionnaire surveys may be low and differ between cohorts, introducing potential bias to findings.

## INTRODUCTION

Injuries cause five million deaths worldwide each year and many more people are left with disability.[1] In the UK around six million Emergency Department (ED) visits result from accidental injuries each year[2] and over 14 000 of these injuries result in death.[3]

A founding and sustained principle of the National Health Service (NHS) is that there should be equity of access and treatment for all.[4] However, disparities in access to healthcare and health outcomes for people from ethnic minorities compared with white British people have been regularly reported.[5] Future changes in the delivery of NHS care as proposed in the NHS Long Term Plan[6] may deepen inequalities, as people with urgent care needs including minor injuries are redirected towards NHS 111 (telephone service) and general practitioner (GP) led Urgent Treatment Centres. Following COVID-19, further initiatives have been trialled to control immediate access to emergency care.[7] However, there are concerns that people from ethnic minorities are more likely to make

greater use of emergency healthcare, reflecting difficulties in accessing primary care.[8] In the Health Experiences of Asylum Seekers and Refugees (HEAR) study,[9] 77% of survey respondents knew about the 999 service, but only 28% were aware of the Out of Hours GP service. Research across Europe[10] reports a rise in migrants' and asylum seekers' use of emergency services. High use has been associated with language barriers, social deprivation, poor access to primary care,[10] delayed or restricted access to secondary healthcare[11] or people falling through gaps between other services (such as community mental health services).[12]

People from ethnic minorities across Europe, North America and Oceania have been widely reported to have differences in access, experiences and outcomes when presenting to emergency services. People from ethnic minorities who present with injuries have different experiences in relation to pain management,[13] length of hospital stay,[14] quality of care,[15] disability,[16] repeat attendance[17] and mortality.[18] They also have increased risk of certain injury presentations including gunshot injuries,[19] long bone fractures,[20] head injuries,[21] alcohol-related injury,[22] workplace injury,[23] assaults,[24] self-harm and attempted suicide[25] and Female Genital Mutilation.[26] However, people from ethnic minorities have lower prevalence of other injuries including: falls among the elderly and road traffic injuries,[27] fire injuries[28] and partner violence.[29]

While death and morbidity rates due to injury are higher in some ethnic minority populations in the UK,[30] there remains a gap in evidence on their experiences of emergency services. This is partly due to a lack of focus or priority on this area of inequality until recently.[31] First, there is a weakness in routine information systems, where ethnicity data are often poorly recorded, particularly in emergency prehospital care settings.[32] Second, the preferred language of patients from ethnic minorities are not recorded in routine health data, nor are differences in culture and language adequately accommodated for in emergency services, with a scarcity of government-funded interpreters,[33] public health campaigns[34] and allied health services.[35] There is considerable scope for taking a more analytical approach to studying injury presentation and differences in emergency care among people from ethnic minorities in the UK that will inform policy and practice and help to reduce future disparities and burden of injury, mortality and disability.

## Study aim

To describe disparities in injury presentation, processes of care and outcomes between people from ethnic minorities and white British people when they contact emergency health services for injury.

## Objectives

We will:

1. Describe the published literature reporting all-cause mortality of people presenting with injury to emergency services by ethnicity.
2. Describe the quality (completeness, consistency) of ethnicity data in routine emergency healthcare datasets.
3. Compare between people from ethnic minorities and white British people: injury type, severity, care delivered, outcomes, beliefs and experiences when they contact emergency health services for injuries.
4. Explore with people from ethnic minorities, including refugees and asylum seekers: knowledge of service availability, factors which deter or encourage them to seek help, experiences of emergency healthcare for injuries.
5. Explore emergency healthcare providers' experiences of delivering care to people from ethnic minorities presenting with injury.
6. Synthesise quantitative and qualitative findings to:
   a. Help policy makers and care providers to develop and implement interventions to promote accessibility of services for injury in ethnic minorities populations.
   b. Enable ambulance service and EDs to improve care and outcomes for people in these populations with injuries.
   c. Inform injury surveillance resources to include ethnicity in their reporting of injury.

## METHODS

### Setting

We will conduct this study in the catchment area of one receiving hospital ED within each of four ambulance services (table 1). We selected sites where an established electronic patient data capture system was in place in the ambulance service. The participating ambulance services will provide linkable electronic datasets including ethnicity codes, which are available in approximately 70%

**Table 1** Study sites and partners

| Ambulance service | Emergency department | Third-sector organisation |
|---|---|---|
| East Midlands Ambulance Service | Leicester Royal Infirmary, University Hospitals of Leicester NHS Trust | The Race Equality Centre |
| South East Coast Ambulance Service | East Surrey Hospital, Surrey and Sussex Healthcare NHS Trust | Surrey Minority Ethnic Forum |
| Scottish Ambulance Service | Royal Infirmary of Edinburgh, NHS Lothian | The Welcoming |
| Yorkshire Ambulance Service | Northern General Hospital, Sheffield Teaching Hospitals NHS Foundation Trust | Refugee Council |

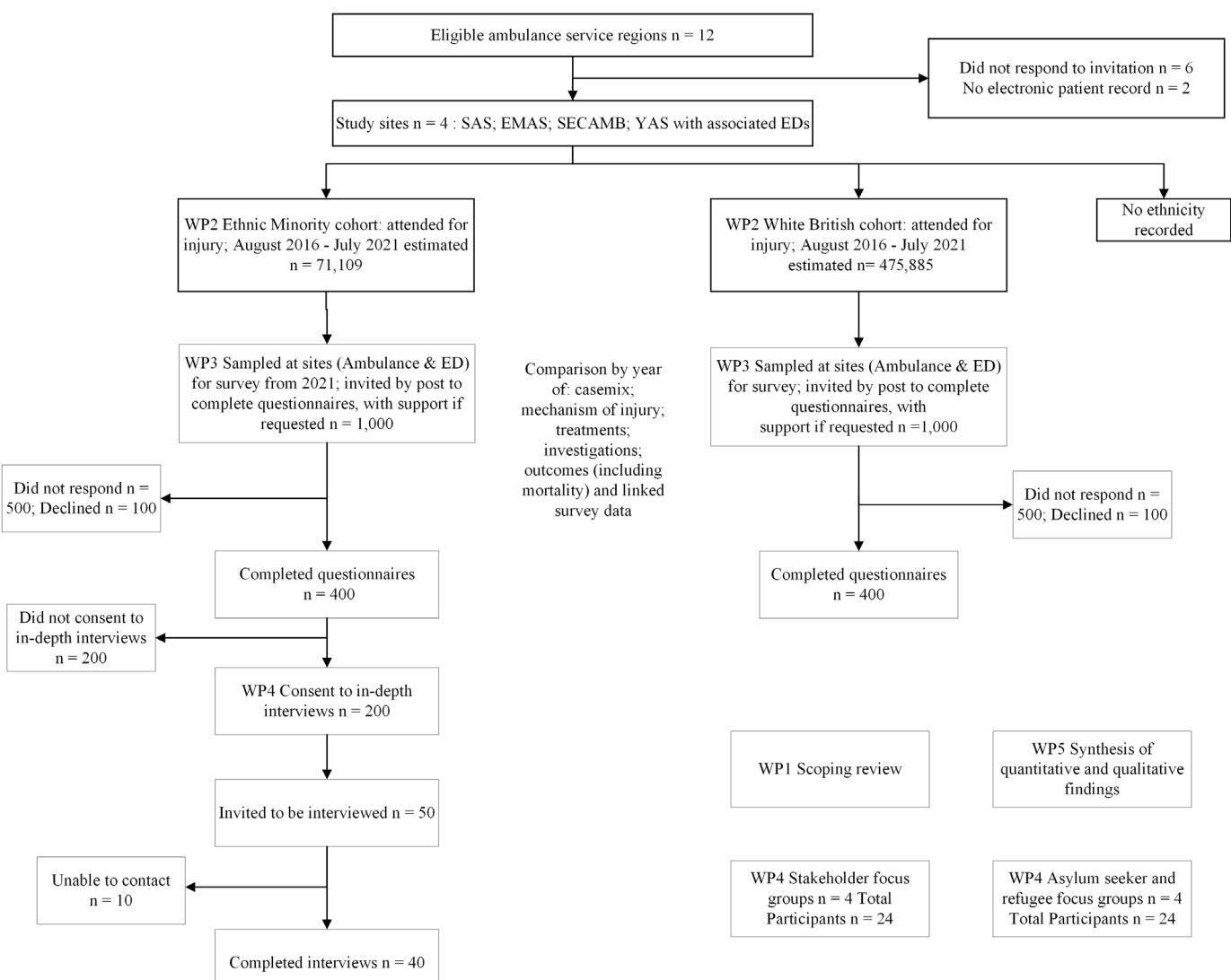

**Figure 1** Study design and participant recruitment flowchart. ED, Emergency Department; WP, Work Package.

of patient records.[32] We hope to retrieve 80% of centrally held ED records with ethnicity codes.[36] The study will begin on 01/10/2021 and end on 30/09/2023.

Third-sector organisations that provide support in relation to healthcare for people from ethnic minorities at each site will help connect researchers with the local population. They will promote the study across their networks to encourage people to respond to postal survey questionnaires and provide support with recruiting and managing peer researchers to support sampled patients to complete the questionnaires.

### Study design

We will use a convergent model of data collection where the quantitative and qualitative methods are conducted in parallel, and results are merged during the interpretation stage.[37] We will examine disparities in experiences, injury presentation, processes of care and outcomes as both the product of the individual patients' decisions and actions[38] and the organisation decisions, actions and attitudes.[39] As shown in figure 1, in our Work Packages (WP), we will:

WP1: Conduct a scoping review of existing literature.

WP2: Retrieve and analyse retrospective linked NHS routine data over a 5-year period (2016–2021) related to ambulance and ED contacts by patients from ethnic minorities and white British patients for injury to compare demographics, casemix, processes and outcomes of care.

WP3: Conduct a questionnaire survey with samples of people from ethnic minorities and white British people who contacted the ambulance service or attended ED for an injury within a specified recent period of up to 6 months to compare self-reported experiences, satisfaction and health-related quality of life.

WP4: Conduct in-depth interviews in each site with people from ethnic minorities who consent to be contacted for an interview in their completed questionnaires and conduct one focus group with refugees and/or asylum seekers at each site and one focus group with stakeholders at each site: for example, ED clinicians, paramedics, GPs and other primary care staff, social services staff, and third-sector support workers.

WP5: Synthesise our findings from quantitative and qualitative data to generate key messages and implications for policy and service delivery.

## WP1: Scoping review

We will undertake a scoping review following the Joanna Briggs Institute (JBI) methodology.[40] We will refer to the Preferred Reporting Items for Systematic Reviews and Meta-analysis Extension for Scoping Reviews (PRISMA-ScR)[41] and JBI reporting checklists to developing a scoping review protocol.[42] The scoping review will aim to describe the published literature reporting cases of mortality by race or ethnicity of adults presenting with injury to emergency services.

We will identify papers from database searches (EBSCO=CINAHL, MEDLINE and PsycInfo; SCOPUS and COCHRANE) which will be screened independently by title, abstract and full paper following a protocol by at least two reviewers from the research team (see online supplemental file 1). We will include studies that report all cases of mortality by race or ethnicity of adults presenting to emergency services for injury. We will exclude studies involving people with no ethnicity record; no record of injury as the cause of emergency service use; no reporting of mortality by race and ethnicity and those reporting non-emergency care such as scheduled appointments, outpatient department services and elective surgeries. All discrepancies between reviewers will be resolved by a third reviewer.

## WP2: Epidemiology of injury presentation, care delivery and outcomes using anonymised linked routine data

We will link routine ambulance service data between August 2016 and July 2021 related to patients presenting with injury within the ambulance service catchment area of each participating ED to centrally held ED, inpatient, outpatient and Office of National Statistics (ONS) datasets (using NHS Digital in England and eDRIs in Scotland).[43] Individual-level prehospital data on calls made for injury will be retrieved by each ambulance service from its computer-aided dispatch and patient clinical record systems; these data are currently unavailable in NHS Digital or Electronic Data Research and Innovation Service (eDRIS). Clinical data will include ethnicity; condition code; job cycle time (from first 999 call for the incident to time ambulance reported free to respond to next 999 call); medications given and disposition (conveyed to hospital, treated without conveyance). We will link this data using a study-specific Identity to patient-identifiable data held in separate files—the 'split file' method[44]—and uploaded to NHS Digital or eDRIS by each site. We will then use patient-identifiable data within NHS Digital or eDRIS to create anonymised linkage fields and retrieve routinely recorded outcomes for these patients (figure 2).

We will also retrieve routine data on ED attendances for injury from participating EDs for the same period from NHS Digital/eDRIS. We will then retrieve anonymised linked routine health outcomes for 6 months after index presentation with injury to ambulance services and EDs to assess outcomes unless the person has specifically opted out.[45] We will request data related to: diagnoses; disposition from ambulance service and ED; length of stay at index episode in hospital; treatments received and discharge code; Intensive Care Unit (ICU) admissions and length of stay; further ED attendances and emergency admissions and deaths up to 6 months.

We will partition the aggregated data into cohorts of patients from ethnic minorities, patients from the white British population and those for whom no ethnicity is recorded, with appropriate subgroups identified using 2011 Census ethnicity categories.[46] We will include patients with multiple presentations or attendances with the first presentation or attendance as baseline and data from subsequent presentations or attendances contributing to outcomes. We will hold a consultation workshop with stakeholders at the outset of the study to help clarify and define our study outcome measures.

We will compare patterns of presentation, processes of care and outcomes through cross-sectional analyses to investigate differences in:

- ► Demographics; geography and deprivation index; mechanism of injury; severity; injury type (accidental, non-accidental, assault, self-harm); casemix; route to care (direct, via 111 telephone advice service or via general practice).
- ► Treatments and investigations.
- ► Potential safety incidents (eg, hospital admission or death within 72 hours of discharge from 999 or ED care) following injury.
- ► Immediate outcomes (at index event) including ambulance attendance, transportation to hospital, hospital admission, length of stay in hospital and ICU, death following injury.
- ► 6-month outcomes (further ED or hospital attendances, length of stay in hospital and deaths) following injury.

## WP3: Questionnaire survey

We will survey people from ethnic minorities and white British people who have presented with injury to one of the four ambulance services or nominated ED. Each of the four study sites will search through their routine ambulance service and ED records to identify patients presenting with injury and coded as being from an ethnic minority, including patients who presented to the ambulance service but were not conveyed to hospital. They will also identify a similar-sized cohort of patients identified as white British.

Each study site (comprising one ambulance service and one ED) will send out 500 postal questionnaires (n=2000 in total, 4 sites) to 250 patients from ethnic minorities and 250 white British patients. Before sending out questionnaires, the clinical care team will check death records to ensure that the person has not died to avoid causing distress to their family. All recipients will be asked to

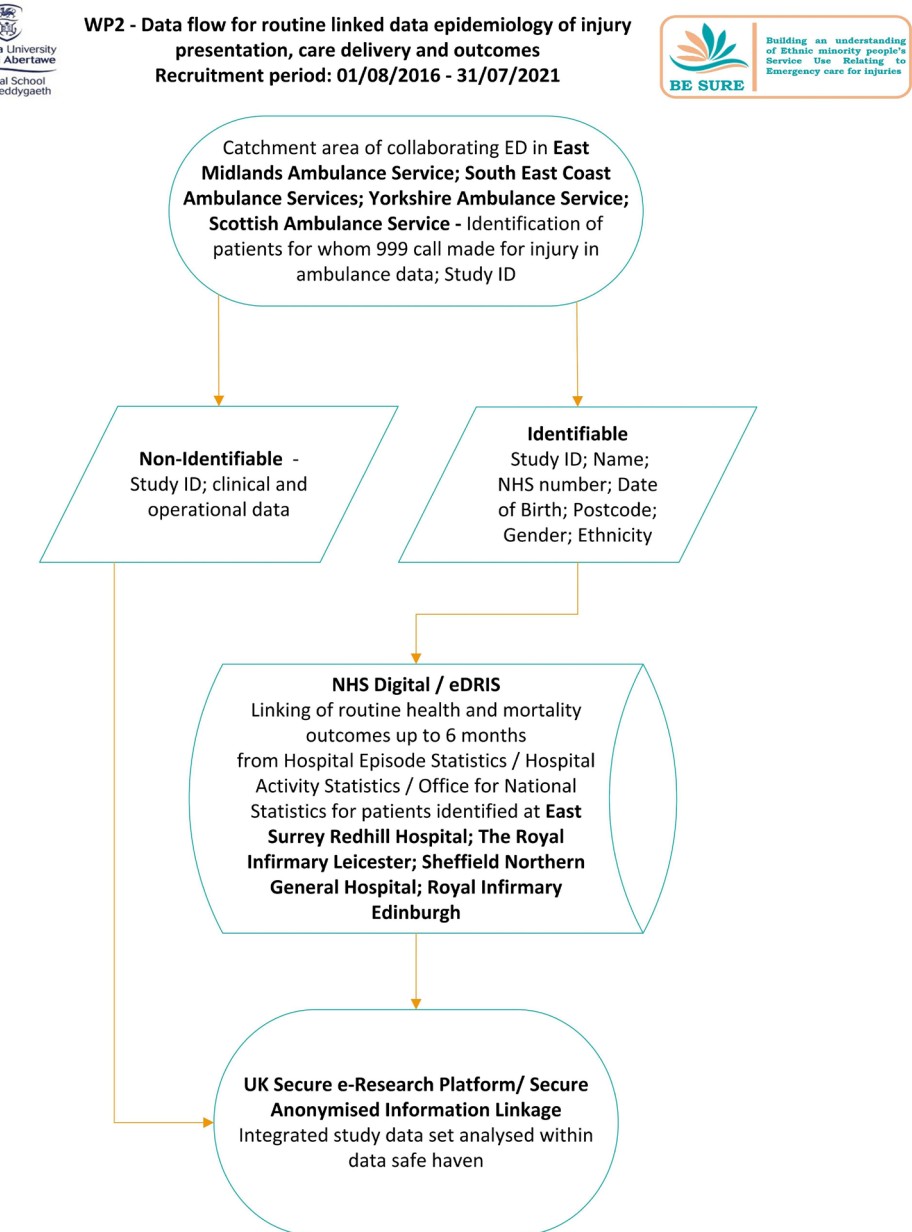

**Figure 2** WP2—Data flow for routine linked data epidemiology of injury presentation, care delivery and outcomes.

return completed questionnaires in a prepaid envelope to Swansea University. A reminder letter will be sent after 2 weeks. Recipients will also have the option to complete the questionnaire online (via a secure approved platform) using a QR code, reducing the potential burden of having to return the questionnaire by post. Where patient contact numbers are available, the clinical care team will contact the patient by telephone 1 week after sending the questionnaire, to offer support to complete the questionnaire over the telephone or to refer the participant to a local peer researcher to provide this support where consent is provided. The expected 800 analysable questionnaire responses will (using 90% power, 5% significance) enable us to detect differences in outcomes equivalent to a standardised statistical effect of ~0.23; this, in turn, corresponds to clinically meaningful

differences in study outcomes (eg, self-reported Health Related Quality of Life). We will offer all respondents a £10 voucher for completing the questionnaire.[47] All questionnaire data will be inputted and stored on secure Swansea University database.

We will base our survey questions on those used successfully in the HEAR survey,[9] focusing on knowledge of services, beliefs, experiences of injury, expectations and health-seeking behaviour. The questionnaire (see online supplemental file 2) will also include standardised questionnaires to measure satisfaction with care (Quality of Care Monitor)[48] and current health status (SF-12).[49] The questionnaire will be translated into several languages and translated versions will be available on request.

We will recruit and train 12 community peer researchers[50] from ethnic minorities to support with

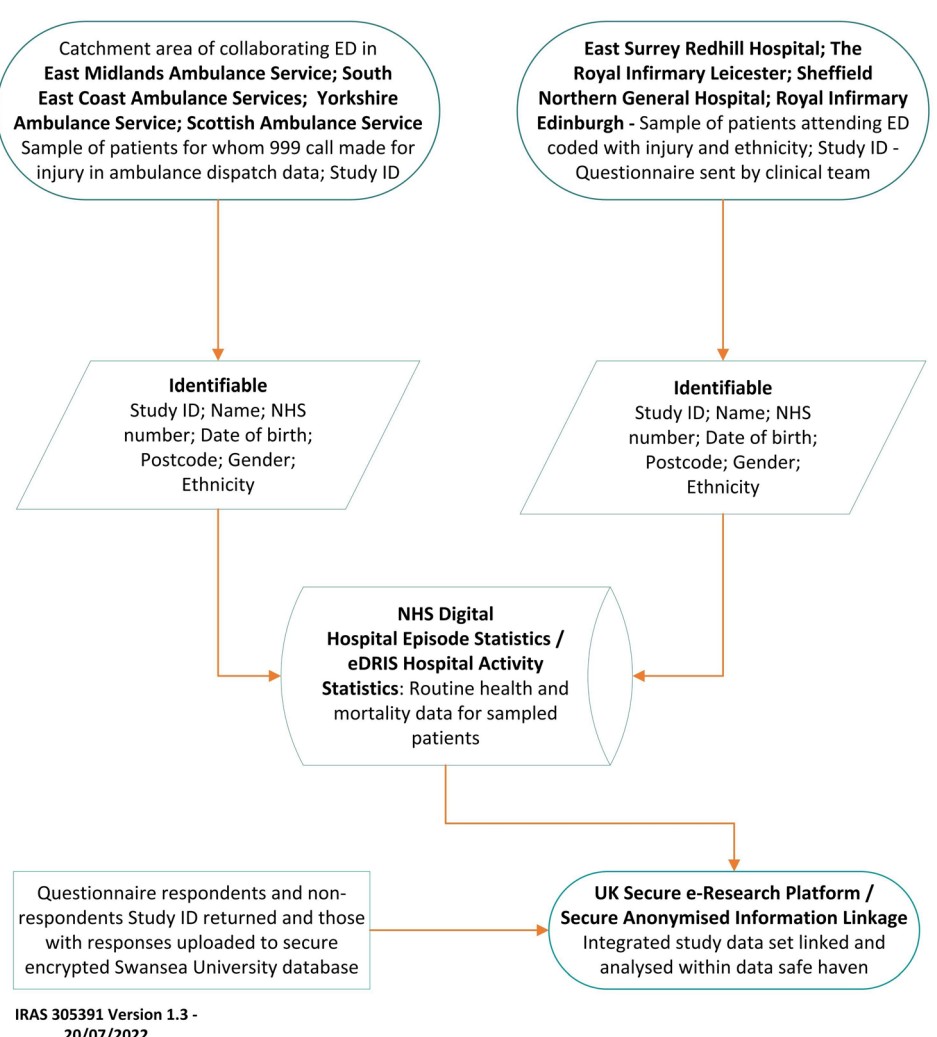

**WP3 - Data flow for survey**
**Recruitment period: Patients presenting with injury**
**Dates to be confirmed following HRA approval**

Catchment area of collaborating ED in **East Midlands Ambulance Service; South East Coast Ambulance Services; Yorkshire Ambulance Service; Scottish Ambulance Service** Sample of patients for whom 999 call made for injury in ambulance dispatch data; Study ID

**East Surrey Redhill Hospital; The Royal Infirmary Leicester; Sheffield Northern General Hospital; Royal Infirmary Edinburgh -** Sample of patients attending ED coded with injury and ethnicity; Study ID - Questionnaire sent by clinical team

**Identifiable** Study ID; Name; NHS number; Date of birth; Postcode; Gender; Ethnicity

**Identifiable** Study ID; Name; NHS number; Date of birth; Postcode; Gender; Ethnicity

**NHS Digital Hospital Episode Statistics / eDRIS Hospital Activity Statistics**: Routine health and mortality data for sampled patients

Questionnaire respondents and non-respondents Study ID returned and those with responses uploaded to secure encrypted Swansea University database

**UK Secure e-Research Platform / Secure Anonymised Information Linkage** Integrated study data set linked and analysed within data safe haven

IRAS 305391 Version 1.3 - 20/07/2022

**Figure 3** WP3—Data flow for questionnaire survey.

the collection of questionnaire data. We will recruit and train peer researchers with support from our third-sector partner organisations at each site. The peer researchers will work closely with the clinical care team in their localities who will refer respondents who request language support to the appropriate peer researcher with the patient's consent to help complete the questionnaire.

Identifiable data for all patients who are sampled to receive the questionnaire will be saved by participating services in a 'split file' format and uploaded into NHS Digital or eDRIS (figure 3). Questionnaire responses will be linked to clinical records from ED (Hospital Episode Statistics (HES) and Emergency Care Data Set (England); Hospital Activity Statistics and Ambulance & Emergency Datamart (Scotland)) and hence to factors and covariates derived from these data sources.

### WP4: Qualitative interviews and focus groups

We will conduct a total of 40 interviews with patients from ethnic minorities (10 in each site, identified from survey respondents who provide their consent and contact details) and four focus groups with Asylum Seekers and Refugees (up to six participants per site, identified by third-sector organisations).

We will purposively sample survey respondents who have experienced an injury in the previous 12 months by key characteristics such as injury type, injury severity, age, gender and ethnic background to provide consent for interview (see online supplemental file 3). Where language is a barrier, an interpreter will be present. Those who participate will be provided with contact numbers for support groups should they experience any distress during their participation and would like additional support. We will offer all participants in patient interviews and focus groups a £20 voucher in recognition of their

contributions.[42] We will conduct all focus groups at the premises of a local third-sector organisation.

We will also conduct four focus groups with stakeholders, one in each site with up to six participants, supplemented with telephone interviews as necessary. We will recruit stakeholders from a range of professional groups: ambulance call takers, paramedics and operational managers; ED clinicians; GPs and other primary care staff; social services staff and third-sector support workers. We will explore participants' experiences and practicalities of delivering care to patients from ethnic minorities who present with injury, including their resource and training needs. We will offer all participants a £20 voucher to acknowledge the time taken to contribute to the study.[51]

### WP5: Synthesis of quantitative and qualitative findings

We will synthesise findings obtained from the four previous WPs to ensure that findings assist policymakers, commissioners and care providers to achieve the best outcomes for patients, staff and the wider healthcare system.

### Analysis

#### WP1: Scoping review

We will chart data reporting on aims, sample size, demographics, injury presentation, cause of injury, definition of mortality and the difference in rate of mortality. We will describe but not appraise included papers for methodological quality or risk of bias, which is consistent with guidance on conducting scoping reviews.[42]

#### WP2: Epidemiology of injury presentation, care delivery and outcomes using anonymised routine linked data

Our statistical analysis plan will characterise and allow for differences in population between study sites. We will interpret results in light of these differences to maximise generalisability across the UK population. We will detail conventions on comparison of processes and outcomes (including inclusion and exclusion rules for covariates and factors), management of missing data, selection of confounders and the reporting of outcomes. To ensure we can report on outcomes by ethnicity (and ethnic subgroups, where appropriate), we will cross-reference and validate key variables across data sources (eg, HES or ONS and CCG (England)). We will adjust our comparisons between cohorts (people from ethnic minorities and white British) and subgroups using prespecified factors and covariates (eg, age; gender; socioeconomic status) obtainable from routine data. We will request deprivation measures associated with patient residence. These socioecological data will comprise an Index of Multiple Deprivation and component domains and we will include appropriate summaries as confounders in our statistical models.

We will describe and compare when analysing our routine data, summarising the epidemiology of injury by ethnicity (including patterns of presentation; injury type, severity and case-mix; processes and outcomes of care)

based on those presenting to the emergency services within and between people from ethnic minorities and white British people. We will include analysis by ethnic subgroups where numbers allow. Across the four study sites, we expect to identify approximately 70 000 people from ethnic minorities and 480 000 white British people who have sought emergency care for injury. This will give ample power to undertake meaningful comparisons across aspects of presentation (eg, proportion presenting with a specific condition), disposition (eg, proportion admitted to hospital; length of stay) and further outcomes (eg, reattendance rates, mortality) over time and between cohorts and prespecified subgroups.

Limitations in routine data will define a third study cohort, comprised of people presenting with injury but for whom no useable data on ethnicity are available. We will describe the characteristics and outcomes (eg, age, sex, injury type and severity; and health outcomes) for this cohort and compare them with the people from ethnic minorities and white British cohorts. This will address our objective related to the quality of ethnicity data in emergency care settings.

#### WP3: Questionnaire survey

We will collate questionnaire data on a secure platform; initial processing will include data validation, assessment of its quality and completeness and implementation of published scoring algorithms. In our analyses, we will report descriptive summaries of responses (using standard methods, including tabulated counts and percentages); comparative analyses, combining questionnaire outcomes with prespecified factors and covariates and, where feasible, description and comparison of questionnaire respondents and non-respondents.

#### WP4: Qualitative interviews and focus groups

We will use framework analysis[52] to analyse qualitative interview and focus group data. We will identify themes from our study questions, the literature and initial analysis of survey data to develop our framework. We will code transcripts according to these themes and refine as analysis progresses. Experienced qualitative researchers will lead analysis of interview and focus group transcripts. Two public contributors will help to validate the analysis process, supporting key stages of coding, refining themes and providing a critical stance.[53] We will use NVivo 11, computer-assisted qualitative data analysis, to manage data. We will remove all identifiable data from interview and focus group transcripts and assign a participant number for identification. Where appropriate, anonymous coded excerpts or quotes will be included in outputs.

#### WP5: Synthesis of quantitative and qualitative findings

We will synthesise and report on quantitative and qualitative findings by identifying meta-themes that cut across each component of the study.[54] We will interpret the results and consider similarities and differences, including

recurring themes and issues that emerge from the scoping review, routine data, survey responses and people's views and experiences of injury and care received. We will use this evidence to inform our policy recommendations for improving injury care for people from ethnic minorities, including direction of future research.

## ETHICS AND DISSEMINATION

We have obtained a favourable ethical approval from the Wales Research Ethics Committee (305391). We have also completed all necessary research permissions through the Health Research Authority. In addition, we obtained information governance approvals from the Confidentiality Advisory Group to conduct data linkage and retrieval of outcomes for analysis from NHS Digital in England and are in the process of gaining approval from eDRIS in Scotland. Due to data protection and patient confidentiality, participating Trusts are unable to share medical records with peer researchers or third-sector organisations. Therefore, the research paramedics and nurses will identify and recruit participants from routine records to take part in the questionnaire survey (WP3) and qualitative interviews (WP4).

### Patient and public involvement

We will ensure our public contributors are actively involved in all aspects of the study.[55] We have strong relationships with people from ethnic minorities who have contributed experience-based expertise throughout the process of planning this proposal. We have drawn on the experiences and knowledge of two experienced public contributors to design the study who will join the Research Management Group to implement the research (TH, SS). We will recruit two additional public contributors to join the independent Study Steering Committee alongside clinical, policy, academic, methodological and subject experts. We will also regularly present progress and emerging findings of our study to two public advisory groups, the PRIME SUPER Group[56] and the SAIL Consumer Panel.[57] We will provide honoraria, briefings and other support as needed in line with best practice and report public involvement in our outputs.[58]

### Dissemination

We will include engagement with patient and professional groups, NHS managers, commissioners and policy makers and third-sector organisations in our communication, publication and dissemination plan. We will use the plan to guide our second Stakeholder Event, which will take place once the study data collection and analysis are complete. The Stakeholder Event will be designed to be inclusive allowing patients, public contributors, third-sector organisations, service providers and policy makers the space to share their views. At the event, we will discuss and refine our findings to ensure our results are credible and are widely shared with the community and service providers.

## DISCUSSION

This is the first study in the UK to use routine anonymised linked data to compare outcomes and experiences for people from ethnic minorities and white British people when they present with injury to emergency health services. Our mixed-methods design builds on this innovative approach to capturing data by employing qualitative methods (WP4) to gain an in-depth understanding of a range of experiences, outcomes and views about emergency care that are not available in routine health records. Our focus groups with asylum seekers and refugees (WP4) provide a valuable insight into the ways an already vulnerable population accesses and navigates emergency health services when contacted for care following injury.

The strength of this study lies in its multifaceted approach to study design, data collection and analysis which stems from the diverse study team. Collaborating with ambulance services, EDs, community members and third-sector organisations strengthens the implementation of the study's research activities and ensures that the contribution this study makes to the evidence base will be informed by those who deliver and use emergency services.

**Contributors** FBa drafted the manuscript with editorial input from all authors. FBe, BAE, SG, TH, AK, AJ, PAJ, RL, AP, ITR, SS, ANS, AW, JW, HS. The research idea was conceived by AK and HS and developed by all authors. All authors read and approved the final manuscript.

**Funding** This work was supported by National Institute for Health and Care Research grant number NIHR132744.

**Competing interests** None declared.

**Patient and public involvement** Patients and/or the public were involved in the design, or conduct, or reporting, or dissemination plans of this research. Refer to the Methods section for further details.

**Patient consent for publication** Not applicable.

**Provenance and peer review** Not commissioned; externally peer reviewed.

**ORCID iDs**
Fadi Baghdadi http://orcid.org/0000-0002-3770-7611
Bridie Angela Evans http://orcid.org/0000-0003-0293-0888
Steve Goodacre http://orcid.org/0000-0003-0803-8444
Ann John http://orcid.org/0000-0002-5657-6995
Ronan A Lyons http://orcid.org/0000-0001-5225-000X
Alison Porter http://orcid.org/0000-0002-3408-7007
Aloysius Niroshan Siriwardena http://orcid.org/0000-0003-2484-8201
Alan Watkins http://orcid.org/0000-0003-3804-1943

Julia Williams http://orcid.org/0000-0003-0796-5465
Ashrafunnesa Khanom http://orcid.org/0000-0002-5735-6601

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
