## [Reviewer comments · BMJ Open]

ARTICLE DETAILS

TITLE (PROVISIONAL)	Building an understanding of Ethnic minority people's Service Use Relating to Emergency care for injuries: the BE SURE study protocol
AUTHORS	Baghdadi, Fadi; Bell, Fiona; Evans, Bridie; Goodacre, Steve; Hettiarachchi, Thanuja; John, Ann; John, Paul; Khanom, Ashrafunnesa; Lyons, Ronan; Siriwardena, Aloysius; Porter, Alison; Russell, Ian; Safari, Solmaz; Watkins, Alan; Williams, Julia; Snooks, Helen

VERSION 1 – REVIEW

REVIEWER	Pan, Daniel University of Leicester
REVIEW RETURNED	12-Dec-2022

GENERAL COMMENTS	Thank you for asking me to review this manuscript. It is an important topic to address and the study design is well conceived. I believe it deserves publication. Get on and get the study done!
--

REVIEWER	Eaton, Georgette University of Oxford, Nuffield Department of Primary Care Health Sciences
REVIEW RETURNED	26-Feb-2023

GENERAL COMMENTS	The abstract succinctly captures the intended research, which is outlined in detail in the remainder of the article. Each work package is clearly articulated, in accordance with relevant frameworks when required, and the proposed analysis is suitable for the data collected. I note one minor point within the methods section for WP3: It appears that a reference to the HEAR survey is missing (page 11, line 53). As well as the multidisciplinary research team, the approach to patient and public involvement within this protocol (and the research) is exemplary. I have no further comments or recommendations. I look forward to seeing the results of this research when completed, as this promises to be a worthy (and much needed) addition to the evidence base.
--

VERSION 1 – AUTHOR RESPONSE

Response to Reviewer Feedback

Building an understanding of Ethnic minority people's Service Use Relating to Emergency care for injuries: the BE SURE study protocol

We would like to thank the reviewers for the wonderful feedback and very encouraging words – thank you :). Please see below the responses to the editor feedback:

1. **Along with your revised manuscript, please include a copy of the PRISMA-ScR checklist for scoping reviews indicating the page/line numbers of your manuscript where the relevant information can be found**
<https://eur03.safelinks.protection.outlook.com/?url=https%3A%2F%2Fwww.equator-network.org%2Freporting-guidelines%2Fprisma-scr%2F&data=05%7C01%7Cfadi.baghdadi%40swansea.ac.uk%7C6422bfed6a5948637a4a08db1a3eb602%7Cbbcab52e9fbe43d6a2f39f66c43df268%7C0%7C0%7C638132628859433524%7CUnknown%7CTWFpbGZsb3d8eyJWljojMC4wLjAwMDAiLCJQIjoiV2luMzliLCJBTil6Ik1haWwiLCJXVCi6Mn0%3D%7C3000%7C%7C%7C&sdata=89TLN7%2BGhee5idCn%2BxIkCaSKaWdAFLv68QfUGK3bpbl%3D&reserved=0>. As your study is a Protocol, not all of these points will be relevant, please indicate "Not applicable" for those.

We have attached a PRISMA-ScR checklist for the protocol.

2. **Please upload a copy of the questionnaire and interview guide as Supplementary Material files.**

Uploaded as Supplementary Material files.

3. Please indicate in the Methods section which databases will be searched for the Scoping Review. Please also include, as a supplementary file, the precise, full search strategy (or strategies) for all databases, registers and websites, including any filters and limits used.

- We have included the databases on page 9 in paragraph 1 - EBSCO = CINAHL, MEDLINE and PsycInfo; SCOPUS and COCHRANE
- We have also included a database search strategy as a supplementary file.

4. Please indicate which sites will be included in your study in the Methods section.

We have included a table (see below) on page 7, detailing the sites and partners.

Ambulance service	Emergency Department	Third Sector Organisation
East Midlands Ambulance Services	Leicester Royal Infirmary, NHS Trust	The Race Equality Centre
South East Coast Ambulance service	East Surrey Hospital, Surrey and Sussex Healthcare, NHS Trust	Surrey Minority Ethnic Forum
Scottish Ambulance Service	Royal Infirmary of Edinburgh, History, NHS Lothian	The Welcoming
Yorkshire Ambulance Service	Northern General Hospital, Sheffield Teaching Hospitals, NHS Foundation Trust	Refugee Council

5. Please include the planned start and end dates for the study in the methods section.

We have included the planned start and end date on page 7 - The study will begin on the 01/10/2021 and end on the 30/09/2023.

- 6. Please ensure that all of the information sent to the peer reviewer funders is included in the manuscript, the response "Due to data protection and patient confidentiality, participating Trusts are unable to share medical records with peer researchers or Third Sector organisations. Therefore, the research paramedics and nurses will identify and recruit participants from routine records to take part in the questionnaire survey (WP3) and qualitative interviews (WP4)" appears to be missing from the manuscript.**

We have included the required information on page 7, first paragraph.

1. Confirm files if supplementary:

- We have noticed that you have uploaded the file "IRAS 305391 Questionnaire - BE SURE V1", "BE SURE Interview Guide Ethnic Minority Patients" and "BE SURE - Scoping Review - Database search strategy.odt" under 'supplementary file'. However, we can't see any citation for this file within the main text. If this file needs to be published as supplementary file, please cite it as 'supplementary file' in the main text and upload it in PDF format. Or you can change the file designation into "Supplementary file for Editors only".

I have changed the files to "Supplementary file for Editors only".

2. Figure resolution:

- Please re-upload your Figures and make sure that they have a resolution of at least 300 dpi and 90mm x 90mm of width. Figures in DOCUMENT, EXCEL and POWERPOINT format are not acceptable.

I have adjusted the figures to be over 300 dpi and over 90mm x 90mm

3. Reference citation missing:

- Please review your main document reference number 13 citation is missing. References must be numbered consecutively in the order in which they are mentioned in the MAIN TEXT. Please take note reference citations must begin in the introduction of the main text and not in the abstract section of the paper.

I have fixed the citation and reuploaded a clean and marked copy. There are no reference citations before the introduction.

4. Contributorship statement mismatch:

- Please ensure that your "Contributorship statement" in your main document and ScholarOne submission system is the same.

I have ensured both match.